# An Endoglucanase Secreted by *Ustilago esculenta* Promotes Fungal Proliferation

**DOI:** 10.3390/jof8101050

**Published:** 2022-10-07

**Authors:** Zhongjin Zhang, Jiahui Bian, Yafen Zhang, Wenqiang Xia, Shiyu Li, Zihong Ye

**Affiliations:** Zhejiang Provincial Key Laboratory of Biometrology and Inspection & Quarantine, College of Life Sciences, China Jiliang University, Hangzhou 310018, China

**Keywords:** *Ustilago esculenta*, β-1,4-endoglucanase, pathogenic development, *Zizania latifolia*

## Abstract

*Ustilago esculenta* is a fungus of two morphological forms, among the filamentous dikaryon that can induce the plant stem to expand to form fleshy stem. In order to establish biotrophy with *Zizania latifolia* which belongs to the tribe Oryzeae (Poaceae), *U. esculenta* firstly needs to secrete a bunch of effectors, among them being cell wall degrading enzymes (CWDEs). We have isolated a gene, *UeEgl1*, which was differentially expressed in MT-type and T-type *U. esculenta* at an early stage of infection, and specifically induced in the filamentous growth of the T-type. Bioinformatics analysis and enzyme activity assay indicated that UeEgl1 functions outside the cell as a β-1,4-endoglucanase with a conserved domain of the glycosyl hydrolase family 45 (GH45) which targets the main component of the plant cell wall β-1,4 linked glycosidic bonds. The phenotype analysis of *UeEgl1* deletion mutants and *UeEgl1* over-expression transformants showed that *UeEgl1* had no significant effect on the budding, cell fusion, and filamentous growth of *U. esculenta* in vitro. Further study found that over-expression of *UeEgl1* promoted the proliferation of mycelia inside *Z. latifolia*, and raised plant defense responses. The above results show that the *UeEgl1* gene may play an important role in the early stage of infection through the decomposition of the plant cell wall.

## 1. Introduction

*Ustilago esculenta* is a typical dimorphic plant pathogen. Similar to *Ustilago maydis*, the life cycle of *U. esculenta* exists in two different morphological forms: yeast type and filamentous dikaryon form [1]. Haploid yeast type reproduces by budding and does not have pathogenicity; in this form, two sexually compatible strains fuse to form filamentous binucleates and obtain the ability to infect the only known host *Zizania latifolia* [2]. Its successful infection in *Z. latifolia* can cause edible galls and inhibit flowering [3,4]. With a long-term artificial selection, two types of strains, MT-type and T-type were differentiated [2,5]. The MT-type strains can induce edible fleshy white *Jiaobai*, which is widely planted in Asia as an aquatic vegetable, and selected every year by farmers in the form of asexual reproduction. However, T-type strain invasion leads to the formation of grey *Jiaobai*, which is full of brown teliospores, always discarded by farmers [3]. The MT-type and T-type strains differ in virulence and in causing plant defense responses, with the main apparent difference reflected in the proliferation rate in the plant [6,7]. The T-type strains have higher virulence, and proliferate faster than the MT-type in the host, resulting in a great quantity of teliospores in swollen galls. Instead, the MT-type strains with a limited proliferation only form a few teliospores and are full of fungal hyphae in white *Jiaobai* [7].

As a biotrophy fungus similar to *U. maydis*, *U. esculenta* continuously draws nutrition from the host to proliferate and extend by entering into the plant cell wall, therefore it is necessary to break through the first layer of the plant defense network, the plant cell wall [8,9,10]. There are diverse strategies existed, e.g., hemibiotroph fungi such as *Magnaporthe oryzae* and *Colletotrichum* species possess melanized appressorium with extremely high turgor pressure, which converts to mechanical pressure to help with going through the plant cuticle and plant cell wall [11,12]. Biotrophs such as *U. maydis* and *U. hordei* have non-melanized appressorium and facilitate penetration of the cuticle and plant cell wall by secretion of plant cell wall degrading enzymes (CWDEs) and mechanical force [13,14]. *U. esculenta* is unable to form structures such as appressorium, therefore secretion of CWDEs is most likely the prime strategy to enter into the plant cytoplasm.

Plant pathogens contain a variety of carbohydrate-active enzymes (CAZymes) [15], which are responsible for the synthesis, modification, and break-down of polysaccharide polymer, part of the nutrition source for plant pathogens [16]. More importantly, some of CAZymes degrade complex carbohydrates in plants, to break through the plant cell wall, CAZymes involved in such a process are called CWDEs. CWDEs such as cellulase are closely related to pathogenic virulence [17,18,19,20], GH families of cellulase activity such as GH7 are absent in bacteria, but are common in fungi [21]. Deletion of GH6 and GH7 cellulases in *Magnaporthe oryzae* could not prevent the rapid formation of the papilla in the host, leading to blocked invasion [22]. In addition, CWDEs of different catalytic activity were also reported relating to pathogenicity. All secreted endo-xylanase of *U. maydis* are involved in infection [23]. Pectate lyase (*VdPEL1*) deletion in *Verticillium dahlia* severely compromised its virulence and decreased in inducing both cell death and plant resistance [24]. Knock-out of *PsGH7a* (encoding cellobiohydrolase) of *Phytophthora sojae* also decreased its virulence [25]. Yet, the discovery of the function and related mechanism of CWDEs in fungal pathogenic processes is still rare.

Similar to other Ustilaginomycetes, *U. esculenta* produces a series of mixtures of extracellular CWDEs that may relate to pathogenicity [26]. However, as a biotroph, *U. esculenta* has a significantly lower amount of CWDEs than hemibiotrophy and necrotrophy fungi, and even the other smut fungi [27,28,29]. In this study, we discovered a gene that showed higher expression at multiple stages of the life cycle of the T-type strain, when compared to that in the MT-type. For that phylogenetic analysis and enzymatic activity demonstrated it as a β-1,4-endoglucanase (EG; 3.2.1.4), belonging to glycoside hydrolase family 45 (GH45), so we named it *UeEgl1*. Besides, *UeEgl1* functional properties in the life cycle of T-type strain were comparatively analyzed with *UeEgl1* deletion and over-expression strains, and the possible mechanism of its function in fungal proliferation was further investigated through plant cell wall staining and transcriptome analysis.

## 2. Materials and Methods

### 2.1. Media and Growth Conditions

*Escherichia coli* DH5α (Takara, Japan) was grown in LB medium (peptone 2%, yeast extract 1%, NaCl 2%) at 37 °C. A pair of T-type compatible haploid stains, UeT14 (CCTCC AF 2015016) and UeT55 (CCTCC AF 2015015), were isolated from Longjiao 2# (Variety number: 2,008,024), a variety of grey *Jiaobai*. A pair of MT-type compatible haploid strains, UeMT10 (CCTCC AF 2015020) and UeMT46 (CCTCC AF 2015021), were isolated from white *Jiaobai* of Longjiao 2#. *U. esculenta* were grown in YEPS medium (peptone 2%, yeast extract 1%, sucrose 2%) at 28°C.

### 2.2. Endoglucanase Gene Acquisition

The *UeEgl1* sequence was firstly predicted, by blasting *egl1* of *U. maydis* to *U. esculenta* whole-genome shotgun sequencing (JTLW00000000.1). Then related primers were designed based on the predicted sequence. All the genomic DNA (gDNA) in this research was extracted by the cetyltrimethylammonium bromide (CTAB) method. Total RNA was extracted and purified using a Spin Column Fungal Total RNA Purification Kit (B518659, Sangon Biotech, China), and reverse-transcribed into complementary DNA (cDNA) with HiScript II 1st Strand cDNA Synthesis Kit (+gDNA wiper) (R212-01/02, Vazyme, Nanjing). The genomic sequence and cDNA of *UeEgl1* were amplified by primers egl1-gF/gR and egl1-cF/cR separately (Appendix A). Target sequences were verified by agarose gel electrophoresis and sequencing.

### 2.3. Bioinformatic Analysis

The nucleic acid sequence of *UeEgl1* was analyzed by the softberry website [30] to obtain its predicted open reading frame and amino acid sequence. The gDNA sequence and cDNA sequence of UeEgl1 were compared by clone manager 8 software to obtain intron information. Physicochemical characteristics of UeEgl1 were predicted by the ExPASY website [31]. Pfam [32] and SignalP-5.0 [33] were used to predict conserved domain and transmembrane region, and protein sequences homology of UeEgl1 were compared by the blast function of NCBI website. The phylogenetic tree was constructed by MEGA 6.0 software through the neighbor-joining method. The fungal enzymes in the glycosyl hydrolase family 45 were searched on the CAZy server [34] and UniPort [35], and then were homology aligned with UeEgl1 using the DNAMAN 9 program.

### 2.4. Plasmid and Strain Constructions

*U. esculenta* wild-type isolates UeT14 and UeT55 were used for the generation of all the strains in this study. For the deletion of *UeEgl1*, hygromycin B was used as the resistance selection maker and homologous recombination was chosen as the method [36]. Primers egl1-f-f/R and egl1-r-f/R were used to amplify the sequence fragments of about 1 kb of each flanking region of the *UeEgl1* gene respectively. The sequence contained promoter and hygromycin resistance genes that were divided into two parts with a 25 bp overlapping sequence. The former part of the sequence was amplified by primers Hyg4-F/Hyg-R, and the latter part was amplified by primers Hyg-F/Hyg3-R. Then, two long fragments were obtained by fusion PCR, one of which contained the upstream sequence of the *UeEgl1* gene and hygromycin former part, and the other contained the downstream sequence of the *UeEgl1* gene and hygromycin latter part. They were connected to the pMD19-T-hyg vector (constructed by our lab formerly) with primers egl1-F-F/Hyg3 and Hyg4/egl1-R-R respectively to form plasmids pMD19-T-egl1-F and pMD19-T-egl1-R. Two plasmids were digested by *Kpn* I to obtain the fragments which were together transformed into the protoplast of *U. esculenta* by the PEG3550/CaCl_2_ mediated protoplast transformation method [36,37]. The transformants were screened on the regeneration medium (sorbitol 18.22%, yeast extract 1%, peptone 0.4%, and sucrose 0.4%) with hygromycin resistance at 28 °C for 4–9 days. UeT14ΔUeEgl1 and UeT55ΔUeEgl1 strains were obtained, then verified by PCR amplification and fluorescence quantitative real-time PCR (qRT-PCR) (Appendix A).

For the over-expression of UeEgl1-eGFP, a strong endogenous promoter of *HSP70* was used, and carboxin was chosen as a resistance screening marker [38]. cDNA of *UeEgl1* of UeT14 and UeT55 was amplified by primers egl1-OF with *HSP70* gene homologous arm and primer egl1-OR with *eGFP* gene homologous arm. The amplified sequence was processed under the digestion of the *Nco* I restriction enzyme, and plasmid pUMa932 was under the same digestion. Then, these two fragments were connected by the C112 connection method (Vazyme, C112-01) to construct the pUMa932-HSP-egl1 plasmid. Then linearized by *EcoR* V and transformed to protoplasts of *UeEgl1* deletion strain by PEG3550/CaCl_2_ mediated method. All the transformants were screened on the regeneration medium with carboxin resistance at 28 °C for 4–9 days and verified by fluorescence observation using Leica TCS-SP8 laser confocal microscope (Appendix A). All the primers were listed in Appendix A.

### 2.5. In Vitro Mating Experiment

Strains were cultured in liquid YEPS medium until OD_600_ reached 1.0, then enriched by centrifugation, until OD_600_ value was adjusted to approximately 2.0 in YEPS liquid medium. Sexual compatible haploid *U. esculenta* were mixed in the same ratio and volume. A total of 2.0 μL of the mixed cells suspension was spotted on YEPS solid plate and cultured at 28 °C [2]. Samples were collected every 12 h (h), until 72 h. Conjugation tubes and hyphal growth were observed under XD series optical microscope (Sunnyoptical, Yuyao, China). The colony morphology was observed, scraped under SZN series inverted microscope (Sunnyoptical, China), and collected for subsequent gene expression pattern analysis experiments.

### 2.6. Test for Pathogenicity

As shown in the in vitro fusion experiment, wild-type, *UeEgl1* knockout and *UeEgl1* over-expression strains with OD_600_ value adjusted to around 2.0 were mixed in 1:1 (UeT14 + UeT55 (WT), UeT14ΔUeEgl1 + UeT55ΔUeEgl1 (KO), UeT14ΔUeEgl1::pHSP-UeEgl1-eGFP + UeT55ΔUeEgl1::pHSP-UeEgl1-eGFP (OE)). *Z. latifolia* seedlings at the three-leaf stage were used as infection objects, inoculated on the stem base of the seedlings with a syringe. The cultural condition was 25 °C in the daytime and 22 °C at night, the light cycle was 12 h [7]. Time points at 0, 3, 6, 9, 15, 30, 60, and 90 days were selected for sampling with 3 replicates at each time point.

### 2.7. Gene Expression Analysis

qRT-PCR analysis was performed for the evaluation of gene expression as described [39]. Samples of haploid growth and mating assays were collected on YEPS solid medium at 0–72 h every 12 h. RNA was extracted and transcribed to cDNA. PerfectStart^®^ Green qPCR SuperMix (AQ601, TransGen Biotech) and specific gene primers were used (Appendix A). The process proceeded on a Bio-Rad (CFX Connect™ Real-Time System, Bio-Rad, USA). Relative gene expression was calculated by quantitation-comparative CT (2^−ΔCT^) with *β-Actin* as the internal reference gene [40].

Relative fungal biomass in infected *Z. latifolia* was qualified as described with some modification [41]. Samples were collected at 1 cm above and below the injection site of *Z. latifolia* (including stem tip and part of leaf sheath). Genome DNA was extracted for qRT-PCR analysis. *β-Actin* gene of *U. esculenta* and *Z. latifolia* were analyzed by qRT-PCR with corresponding primers (Appendix A), and relative amounts of *U. esculenta* gDNA were determined by comparative ΔCT of the *U. esculenta β-Actin* gene normalized to the *Z. latifolia β-Actin* gene. All experiments were conducted in three biological repetitions and the error bars depict ± SD, analysis of variance (ANOVA) was performed (*p* < 0.05 indicated significance).

### 2.8. Confocal Microscopic Observation

The infected plant samples were fixed with Carnot fixed solution (absolute ethanol: acetic acid = 3:1) before observation. The sample was washed twice with 1× PBS solution for slicing, and placed in 10% KOH solution for water bath at 85 °C for 0.5~3 h until the slices were translucent. After the completion of hyalinization, removed KOH solution as much as possible by washing with 1× PBS, immersed the sample in fluorescent staining solution (PI 20 μg/mL, WGA 10 μg/mL, CFW 10 μg/mL dissolved in 1× PBS solution) and conducted vacuum filtration (0.04 MPa, 10 min, 3 times) to help to dye. The observation was conducted at 580–617 nm for propidium iodide (PI) excited at 561 nm, 500–540 nm for Triticum vulgaris lectin (WGA) excited at 488 nm, 425–475 nm for calcofluor white (CFW) excited at 405 nm by Leica TCS-SP8 laser confocal microscope, and LAS-AF Lite 4.0 software was used to process image.

### 2.9. Enzymatic Assays

To verify the enzyme activity of UeEgl1, Carboxymethylcellulose sodium (CMC-Na) basic medium (CMC-Na 1.5%, (NH_4_)_2_SO_4_ 0.4%, KH_2_PO_4_ 0.1%, MgSO_4_ 0.05%, and peptone 0.1%) was used to cultivate UeT14, UeT55 and their derivatives at 28 °C. Strains were firstly grown in YEPS medium, collected and enriched strain at OD_600_ value to 2.0, spotted 5 μL on CMC-Na solid medium and CMC-Na liquid medium for 48 h incubation. Then, on the CMC-Na solid medium, 1 g/L Congo red was poured into the plate staining for 30 min, decolorized by 1 mol/L NaCl after rinsing with distilled water. The hydrolysis circle was observed and measured. A total of 0.4 mL supernatant of cell grown in CMC-Na liquid medium were incubated with 0.4 mL of 1% (*w*/*v*) CMC-Na in sodium acetate buffer (0.1 M, pH 5.5) at 50 °C for 2 h. Supernatant added after the incubation was used as a control group. The reduced sugars were measured according to the DNS method [42].

### 2.10. Transcriptome Analysis

Samples collected at 1 cm above and below the injection site after being infected with WT, KO, and OE at 0 and 3 dpi (WT0, WT3, KO3, OE3 with three independent replicates) were used for transcriptome analysis. Total RNA was extracted using RNAprep Pure Plant Kit (QIAGEN, Germany), and RNA integrity was assessed using the RNA Nano 6000 Assay Kit of the Bioanalyzer 2100 system (Agilent Technologies, CA, USA). 

cDNA libraries for RNA-seq were constructed using illumina NovaSeq 6000 (illumine, USA). Index of *Z. latifolia* reference genome (The genome sequence and gene model annotation files of *Z. latifolia* in our lab have yet to be published) was built using Hisat2 and cleaned reads were aligned to the reference genome using Hisat2 [40].

Reads of numbers of genes were counted, and fragments per kilobase of exon model per million mapped fragments (FPKM) of each gene were calculated. Differential expression analysis of two conditions/groups was performed using the DESeq2R package. The *p*-values were adjusted using the Benjamini and Hochberg method. Genes with *p* > 0.5, and absolute foldchange of 2 (log_2_FC) above the threshold of 0 were sorted as “up-regulated”, while genes with a log2 FC below 0 and *p* > 0.5 were “down-regulated”, and genes with *p* > 0.5 were “nonregulated [43]. Cluster Profiler R package to test the statistical enrichment of differential expression genes in KEGG pathways.

## 3. Results

### 3.1. A Predicted Endoglucanase Gene Specifically Up-Regulated during Fungal Filamentation and Proliferation Growth in T-type Strain

In our lab, the transcriptome analysis on the fungal life stage (including invitro budding, invitro mating, and infection) of MT-type and T-type were performed and the data were analyzed to search for genes that differentially expressed between the two type strains at diverse life stage. Among them, a gene (gene ID: g1206) was found with a significantly higher expression in T-type than that in MT-type at 36 and 48 h during the mating stage and 2 and 3 days after inoculation (Appendix A). Generally, this gene was basically not expressed during the budding stage nether in T-type nor MT-type. However, it was up-regulated obviously after 36 h during mating and after 2 days of infection in T-type, but there was an insignificant upward trend and a low level of expression within 48 h during mating and within 3 days during inoculation in MT-type (Appendix A). In the *U. esculenta* genome, g1206 was annotated as endoglucanase 1, with a similarity of 76% compared to *egl1* of *U. maydis*, so named *UeEgl1*.

The expression pattern of *UeEgl1* was further confirmed by qRT-PCR on T-type strains UeT14 and UeT55 (Figure 1A and Figure 2B) and MT-type strains MT10 and MT46 (Appendix A). It was found that the expression of *UeEgl1* was extremely low in all haploid cells. During mating, in T-type cells, *UeEgl1* was merely expressed before 12 h, but up-regulated after 12 h, which was just the mass formation time of conjugation tubes and beginning of the filamentous growth, and reached the peak at 48 h (Figure 1C). In addition, in the infection period, *UeEgl1* was also highly expressed at 3, 6, and 15 days after inoculation, yet, down-regulated at 30, 90 days (Figure 1D). Similar to the transcriptome results, the expression of *UeEgl1* of MT-type in mating and infection stages was still at a low level (Appendix A). This result indicated that *UeEgl1* may function at the early stage of filamentous growth and pathogenic development.

### 3.2. UeEgl1 Encoding β-1,4-Endoglucanse Secreted outside the Cell

According to the *egl1* sequence from *U. maydis* [44], we identified *UeEgl1* (Gene bank accession: OM141134) based on the whole genome sequence database of *U. esculenta* (GenBank: JTLW00000000.1) by local BLAST. Genomic DNA and complementary DNA of *UeEgl1* were amplified and analyzed. *UeEgl1* has a total length of 1112 bp and no intron sequence, encoding a protein that includes 370 amino acids (Figure 2A). Prediction by ExPASY showed a molecular weight of 37.75 KDa and a theoretical isoelectric point of 6.01. Pfam and SignalP-5.0 predicted that *UeEgl1* had a catalytic domain of glycosyl hydrolase family 45 (GH45) located at 27–235 aa (Figure 2A), and a signal peptide, supposing that it is a β-1,4-endoglucanase and has a glycoside hydrolase activity, secrets, and functions outside the cell (Figure 2A).

Phylogenetical analysis of UeEgl1 was carried out with other β-1,4-endoglucanase that have been reported among smut fungi including *U. maydis*, *U. hordei*, *Sporisorium reilianum*, *Sporisorium scitamicum* and *Melanopsichium pennsylvanicum*, and the more distant fungi *Pyricularia oryzae*, *Trichoderma reesei* and *Saccharomyces cerevisiae.* The result shows that UeEgl1 has a high similarity among Ustilaginomycetes, yet, egl1 from smut fungi reveals relatively low similarity to most of the other fungal β-1,4-endoglucanases (Figure 2B). Sequence alignments between the GH45 domain within the smut fungi indicated that endoglucanases of GH45 are highly conserved among smut fungi, and all these five endoglucanase genes comprise signal peptides with high similarity (Figure 2C).

Roughly one-third of the plant cell wall is composed of cellulose, which is connected by β-1,4-glycosidic linkage [45]. *egl1* of *U. maydis* was reported to have cellulase activity [44]. To test whether *UeEgl1* has the ability to catalyze cellulose, wild-type, knock-out mutant, over-expression strains of yeast type and filamentous dikaryon were cultivated on CMC-Na liquid and solid medium. On the solid CMC-Na medium, hydrolytic zones can be observed in the form of yellow circles after being stained and washed. Both haploid and filamentous growth over-expression strains can observe obvious hydrolytic zones. However, neither yeast type nor filamentous dikaryon of wild-type and *UeEgl1* deletion strains detected hydrolytic zones (Figure 3A). DNS method also demonstrated that enzymatic activity of over-expression strains was significantly higher than that of wild-type and *UeEgl1* deletion strains (Figure 3B). *UeEgl1* expression grown on CMC-Na medium at 48 h revealed the same result of enzymatic activity assay that strains grown on CMC-Na medium witnessed high *UeEgl1* expression and enzymatic activity only when *UeEgl1* expressed under strong promoter (Figure 3C). β-1,4 linked glycosidic bonds digestion enzymatic activity of UeEgl1 in CMC confirms that UeEgl1 functions as a β-1,4-endoglucanase which secretes outside the cell.

### 3.3. UeEgl1 Promotes Proliferation of U. esculenta inside Z. latifolia

Highly induced expression of *UeEgl1* after the formation of mycelia suggesting a role in filamentous growth or proliferation in the host. To fully study the function of *UeEgl1* of *U. esculenta*, wild-type, *UeEgl1* deletion and over-expressed strains were assayed. On haploid growth and morphology, neither *UeEgl1* deletion mutants nor *UeEgl1* over-expression strains showed any differences (Figure 4A,B).

On the formation of conjugation tubes or filaments elongation, three pairs of heterogametic strains (UeT14 + UeT55, UeT14ΔUeEgl1 + UeT55ΔUeEgl1, UeT14ΔUeEgl1::pHSP-UeEgl1-eGFP + UeT55ΔUeEgl1::pHSP-UeEgl1-eGFP) were merged and grown on solid YEPS medium. In vitro fusion test showed that the three combinations all can form conjugation tubes at 24 h, and the number was roughly the same. Besides, there was no significant difference in the formation and length of fusion hyphae in each time period. At the same time, they also observed aerial hyphae of the same length and state after 48 h (Figure 4C). Therefore, *UeEgl1* did not play an important role in the mating process of *U. esculenta* in vitro, for it did not affect the development of filamentous growth.

In order to test whether *UeEgl1* affects the invasion of *U. esculenta* into *Z. latifolia* plants, three combinations of WT (UeT14 + UeT55), KO (UeT14ΔUeEgl1 + UeT55ΔUeEgl1), OE (UeT14ΔUeEgl1::pHSP-UeEgl1-eGFP + UeT55ΔUeEgl1::pHSP-UeEgl1-eGFP) were infected the seedlings of *Z. latifolia* growing for two weeks. Observation results of fluorescence confocal microscope showed that on the third day, wild-type strains colonized at the host leaf sheath and began proliferation, while *UeEgl1* deletion strains were still stuck in the invasion phase. Comparatively, *UeEgl1* over-expressed strains were widely distributed in host leaf sheath, showing better colonization and proliferation than that of the wild-type and *UeEgl1* deletion strains (Figure 5A). On the sixth day after infection, the hyphae of all three combinations proliferated in large numbers in plant leaf sheath cells (Figure 5A). A relatively later infection was observed at the stem tip of *Z. latifolia*. There was almost no mycelium observed at 6 dpi. The invading hypha appeared in host stem tips at 9 dpi after WT and OE combinations inoculation, while the hypha of *UeEgl1* deletion strains was observed at 15 dpi. Even more, plants infected with OE combination showed a great number of hyphae evenly distributed in the stem at 15 dpi, yet, plants infected with wild-type and KO combinations had less and sporadic distributed hypha, in which *UeEgl1* deletion strains showed a relatively worse proliferative status (Figure 5A,B). Ultimately, plants inoculated with WT, or KO, or OE combinations all have the ability to form swollen stems containing grey spores (Figure 5C). Furthermore, swollen stems formed by plants inoculated with the OE combination were larger than that of the WT, and swollen stems induced by the KO combination were tinier than that of the WT (Figure 5D). So UeEgl1 acts on the pathogenicity of *U. esculenta*, mainly reflected in promoting the proliferation of *U. esculenta* inside *Z. latifolia*.

### 3.4. Cell Wall Associated Immunity Regulated by UeEgl1 in Z. latifolia

UeEgl1 functions as a cellulase, target on the main component of the plant cell wall. To further confirm its effects on the host cell wall degradation process during infection, we conducted staining on leaf sheath at 3 dpi and stem tip at 15 dpi with WGA/CFW, hyphal growth condition, and changes of the plant cell wall were observed. The result showed that WT and OE reproduced normally in the stem after infection, and the blue fluorescent signal was remarkably weakened in the area with dense mycelial growth compared to the area with less hypha (except for areas with rapid cell proliferation due to fungal invasion). However, hyphal growth of KO was blocked, and the blue fluorescent signal of the plant cell wall was not observed any evident weakening (Figure 6). There was a restriction to the observation of plant stem cells. Therefore, the cell wall fluorescence of infected plants in leaf sheath at 3 dpi was further observed and compared. The result showed that most hyphae of WT and OE can enter into plant cell walls with reduced blue fluorescent signal at the invasion spot. However, partial invasion of KO was blocked with an enhanced blue fluorescent signal at the invasion spot (Figure 6). The above results show that secretion of UeEgl1 of *U. esculenta* contributes to the decomposition of cellulose in plant cell walls, thus invading cells, absorbing nutrients, and achieving proliferation.

After sensing the cell wall damage, plants produce inhibitors such as cellulase inhibitors or express disease resistance genes to cope with the infection of the pathogen [46]. Therefore, we performed transcriptome analysis on the stem tissues at 3 dpi in plants infected by the corresponding strains at 3 dpi (WT3, KO3, OE3). Differentially expressed genes (DEGs) of the *Z. latifolia* infected by WT, KO, or OE were further screened. There are 33263, 32,189, and 32,857 genes differentially expressed in WT3, KO3, and OE3 respectively, with 30,075 commonly expressed in all of them (Appendix A). The clustering heatmap of DEGs showed a diverse regulation pattern in WT3, KO3, OE3, and WT0, especially between KO3 and OE3 (Appendix A). A total of 41,641 predicted genes of *Z. latifolia* were analyzed for their expression patterns. A total of 887 up-regulated genes and 413 down-regulated genes were found between KO3 and OE3 (Appendix A).

Comparison between OE3 and KO3 showed four up-regulated KEGG pathways of the more significant enrichment (red dots) in the host, including MAPK signaling pathway, plant-pathogen, plant hormone signal transduction, and diterpenoid biosynthesis, and two down-regulated pathways, including carotenoid biosynthesis and phenylpropanoid biosynthesis. Between OE versus WT and KO versus WT, down-regulated pathways were mostly the same, and all related to the development of the host and the anabolism of stress resistance-related substances, such as carbon metabolism, carbon fixation, phenylpropanoid biosynthesis, glutathione metabolism, etc. However, there was no significant enrichment pathway at OE versus WT, and only a few genes of circadian rhythm up-regulated at KO versus WT (Appendix A).

It is noteworthy that the significantly up-regulated pathway of plant-pathogen interaction in OE versus KO mainly includes pathogenesis-related (*PR*) genes, antioxidant enzymes, and calcium-binding proteins. These genes were further monitored for their expression difference in the inoculated plants. The *RPM1* (encoding disease resistant protein 1) and the *PR1A* (encoding pathogenesis-related protein 1A) are responsible for recognizing certain products of pathogens and raising disease resistance responses [47,48], the *RBOHB* (encoding NADPH oxidase) regulates reactive oxygen species (ROS) [49], the *CML10* (encoding calcium-binding protein 10) relates to the early stress responses and the *CML49* (encoding calcium-binding protein 49) is thought to regulate cell wall remodeling during the stresses [50,51], most of them were up-regulated in plants infected by *U. esculenta*, indicating a plant defense response of the host to fungal invasion. However, their expression in the OE inoculation group was significantly higher than that of the KO and WT inoculation group at 3 dpi (Figure 7). This expression trend is consistent with the transcriptome result, suggesting a delayed stronger plant defense response in plants after OE invasion.

## 4. Discussion

It is necessary for a combination of different enzymes to degrade the complex structure of plant cell walls, and for fungal pathogens to reach intracellular nutrients [18]. In *U. esculenta*, about 237 fungal CAZys including 46 glycoside hydrolases (GHs) were identified and presumed to play roles in cell wall degradation [27]. Since cellulose is the main component of the plant cell wall, cellulose degradation enzymes including β-1,4-glucosidases, β-1,4-endoglucanases, and β-1,4-exoglucanases are considered promising CWDEs candidates [14]. Studies from many plant pathogens revealed that β-1,4-endoglucanases are highly induced during the development of diseases, such as *egl1* from *U. maydis*, *Plegl1* from *Pyrenochaeta lycopersici* [44,52], yet gene deletion mutant of *U. maydis* and *P. lycopersici* observed no effect on pathogenicity, partly because of redundancy of β-1,4-endoglucanases. There are also phenotype changes caused by β-1,4-endoglucanases knockout mutant, e.g., the deletion of *bglC3* from *X. citri subsp*. Citri showed no extracellular carboxymethyl cellulase activity and delayed infiltration to the host [53]; two of three β-1,4-endoglucanases *eglXoA* or *eglXoB* deletion from *Xanthomonas oryzae* pv. *Oryzae* resulted in a complete virulence-deficiency [54]. Moreover, a double knockout mutant of β-1,4-endoglucanases *MoCel12A* and *MoCel12A* of *Magnaporthe oryzae* observed more severe disease symptoms and greater fungal biomass [55]. Noticeably, although secretion of CWDEs is a common strategy, there are still diverse interaction modes, and differences in enzymes that exert a major role. In *U. esculenta*, the over-expression of *UeEgl1* acts as a cellulase, facilitates its penetration of the host plant cell wall, and promotes its proliferation in the host, but also raised the host defense responses.

Differentiate strains of MT-type and T-type *U. esculenta* have prominent differences in virulence. The MT-type lost its ability to infect *Z. latifolia* from the outside, and its proliferation rate is much slower compared to the T-type [56], eventually forming white *Jiaobai*. The white *Jiaobai* undergoes asexual breeding, so once infected by the MT-type strains, *Z. latifolia* only undergoes asexual reproduction [57], implying that intrusion for *U. esculenta* from outside is no longer needed. When compared MT-type to the combination of KO, they share many similar characteristics, such as low expression of *UeEgl1*, similar phenotype, and partially blocked invasion in early infection. The swollen stem formed by infection of *UeEgl1* deletion strain was slender, indicating a more coordinated growth and development of fungi and host, similar to the appearance of edible vegetables *Jiaobai*. Therefore, in the process of coevolution between *U. esculenta* and *Z. latifolia*, virulence factors such as *UeEgl1* between different varieties of *U. esculenta* have evolved different induced mechanisms after long-term breeding.

In the decomposition process of the plant cell wall by β-1,4-endoglucanase, products such as oligosaccharides were formed, which act as elicitors triggered plant defense responses [17], and β-1,4-endoglucanase itself also is an elicitor [58]. Compare to the wild-type and *UeEgl1* deletion mutant, plant pathways such as MAPK signaling, plant-pathogen, plant hormone synthesis, and signal transduction were up-regulated, and crucial genes responsible for plant defense responses were highly induced by the influence of over-expression of *UeEgl1*, yet fungal proliferation was not suppressed. On the contrary, although there was basically no resistance gene expression in the KO combination, the fungal proliferation was limited. It was speculated that the ability to obtain nutrition was weakened after the degradation of the cell wall was blocked. Compare to the pathogenic T-type, MT-type has an endophytic trend with plant dominated, its ability to trigger plant defense response is weaker [40], yet showed a block in proliferation, similar to *UeEgl1* deletion of T-type. Therefore, the physical barrier of the cell wall may be more important than the resistance induced by the degradation of the cell wall in the interaction between *U. esculenta* and *Z. latifolia*. This weak interaction effect also better protects the endophytic interaction system between *U. esculenta* and *Z. latifolia*.

## 5. Conclusions

In this study, we demonstrated a gene encoding secreted β-1,4-endoglucanase named *UeEgl1*, which is differentially expressed at MT-type and T-type *U. esculenta*, and highly induced when compatible T-type strains fuse and filamentous dikaryon forms. Our result showed that *UeEgl1* mutants and over-expression strains had not revealed any obvious differences in morphology and filamentous growth. However, corresponding to the high expression of *UeEgl1* in the early infection stage, over-expression of *UeEgl1* witnessed a faster proliferation rate inside *Z. latifolia*, with a significantly larger and denser distribution, higher fungal biomass, and bigger swollen stem than that of wild-type and *UeEgl1* deletion strains. Deletion of *UeEgl1* was observed to block both in invading plant cytoplasm and in proliferation. Invasion spots of plant cell wall observed thickened or thinned changes by infection of OE and KO, respectively. Effects of *UeEgl1* deletion and over-expression of *U. esculenta* on plants showed a great change in DEGs; DEGs of pathways related to plant resistance are induced responses to the high expression of *UeEgl1*. All the results indicate *UeEgl1* has an important role in fungal progression inside the plant.

In addition, although hyphal growth can be observed on the CMC-Na medium, *UeEgl1* of wild-type was hardly induced. Low *UeEgl1* expression of wild-type mycelium may be influenced by nutrient deficiency in the culture medium. Afterward, investigation will aim at the mechanism of *UeEgl1* expression, its cooperation with other CWDEs, and the mechanism of triggering plant defense responses. The mechanisms why *UeEgl1* differential expressed at an early stage of infection between MT-type and T-type may be important for us to understand virulence and coevolution between host and pathogen. Furthermore, differential transcriptome screening between T-type and MT-type could be a novel way to identify relative genes.

## Figures and Tables

**Figure 1 jof-08-01050-f001:**
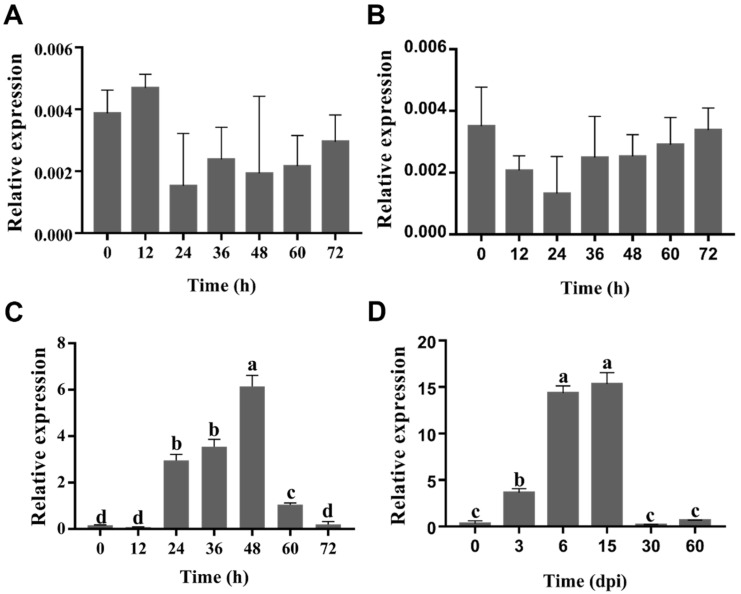
The expression patterns of *UeEgl1* during development stages. (**A**,**B**) *UeEgl1* relative expression of haploid UeT14 and UeT55 strains every 12 h at the budding stage, respectively. (**C**) *UeEgl1* relative expression of T-type strain every 12 h during the mating stage. (**D**) Plant samples infected with merged UeT14 and UeT55 were collected at 0, 3, 6, 15, 30, and 60 days, then *UeEgl1* relative expression was analyzed. The expression of *UeEgl1* was related to the expression of *β-actin*, different letters above the columns represent significant differences at the analysis of variance (ANOVA) (*p* < 0.05).

**Figure 2 jof-08-01050-f002:**
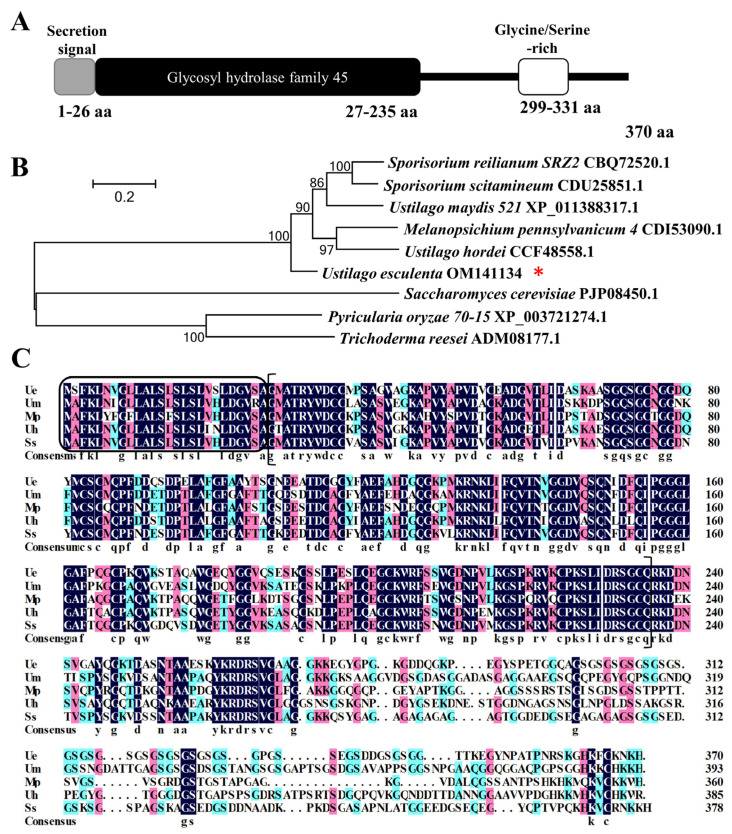
UeEgl1 structure prediction and phylogenetic analysis. (**A**) UeEgl1 comprises 370 aa, grey box (aa 1–26) predicts a putative N-terminal secretion signal, black box (aa 27–235) represents glycosyl hydrolase family 45 (GH45) domain, white box (aa 299–331) is enriched in glycine and serine residues. (**B**) Phylogenetic tree analysis of UeEgl1 with β-1,4-endoglucanase from other fungi. Accession numbers were followed by the name of fungi. *UeEgl1* was marked by “*”. (**C**) Amino acid sequence alignment of GH45 domain. Ue: *U. esculenta*, Um: *Ustilago maydis*, Uh: *Ustilago hordei*, Mp: *Melanopsichium pennsylvanicum*, and Ss: *Sporisorium scitamineum* are chosen for GH45 domain multiple sequence alignment. Identical amino acids are highlighted in black. Amino acids in ellipse are signal peptides, and amino acids in brackets are GH45 domain.

**Figure 3 jof-08-01050-f003:**
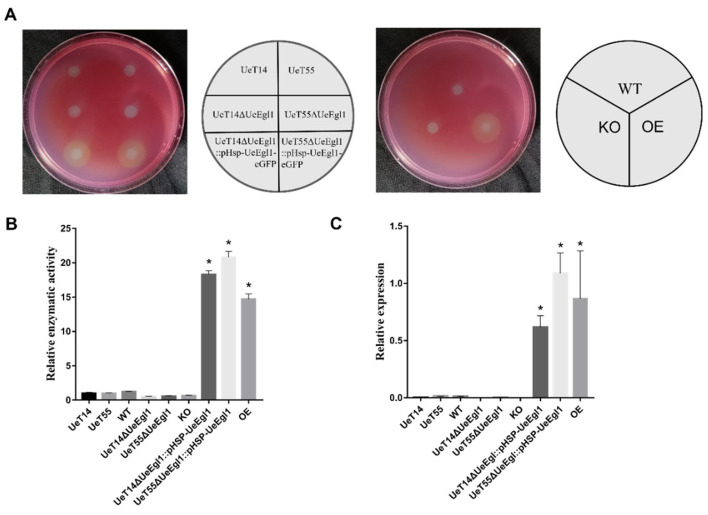
β-1,4-endoglucanase activity of UeEgl1. (**A**) CMC-Na contained in the medium was digested by endoglucanase forming hydrolytic zones, which were yellow circles that existed between the white spot and red background. Names in the right circular correspond to the strains of the left medium. WT is mixed strains of UeT14 + UeT55, KO and OE are strains of mixed UeT14ΔUeEgl1 + UeT55ΔUeEgl1 and mixed UeT14ΔUeEgl1::pHSP-UeEgl1-eGFP + UeT55ΔUeEgl1::pHSP-UeEgl1-eGFP, respectively. (**B**) Relative endoglucanase activity in supernatants of cultures at 48 h. Endoglucanase activity was normalized to that of the UeT14 strain. “*” indicates *UeEgl1* over-expressed strains are significantly higher (Student’s *t*-test, *p* < 0.05) in enzymatic activity than that of the wild-type. (**C**) Relative expression of *UeEgl1* of strains grown on CMC-Na medium at 48 h. “*” indicates *UeEgl1* over-expressed strains are significantly higher (Student’s *t*-test, *p* < 0.05) in enzymatic activity than that of the wild-type.

**Figure 4 jof-08-01050-f004:**
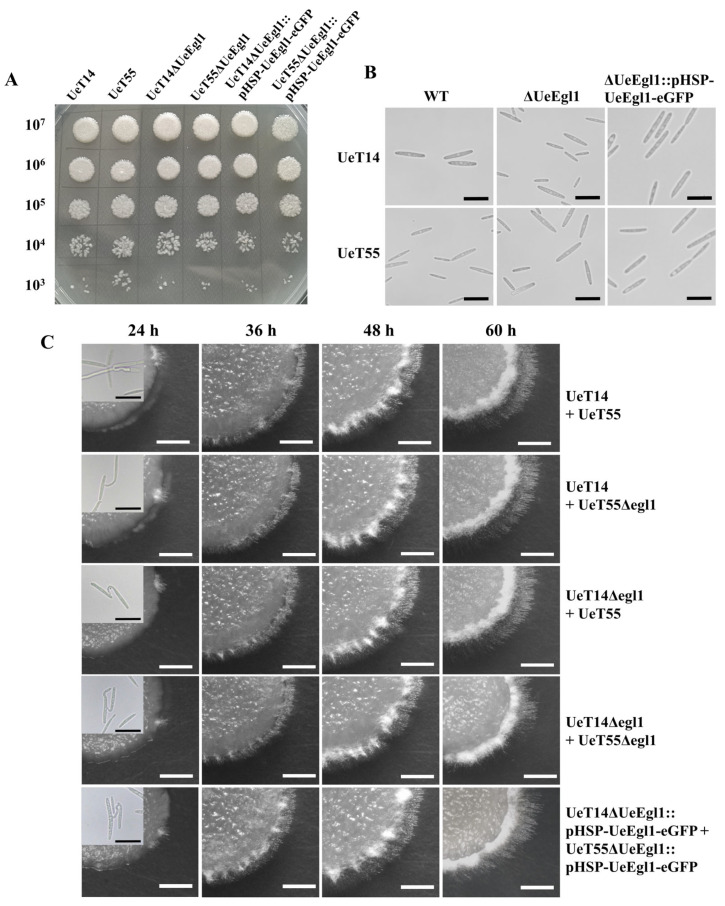
Cells morphology and filamentous growth of *UeEgl1* deletion and over-expression strains. (**A**) The growth rate of haploid wild-type, *UeEgl1* deletion, and over-expressed strains. Strains were spotted on solid YEPS medium at 10^7^, 10^6^, 10^5^, 10^4^, and 10^3^ number of cells. (**B**) Morphology of haploid wild-type, *UeEgl1* deletion, and over-expressed strains. Black bar = 20 μm. (**C**) Morphology of *UeEgl1* deletion and over-expression strains during mating and filamentous growth. Different compatible strains were crossed as above. Images of conjugation tubes were displayed in the top left corner of the strain images at 24 h. White bar = 1500 μm. Black bar = 20 μm.

**Figure 5 jof-08-01050-f005:**
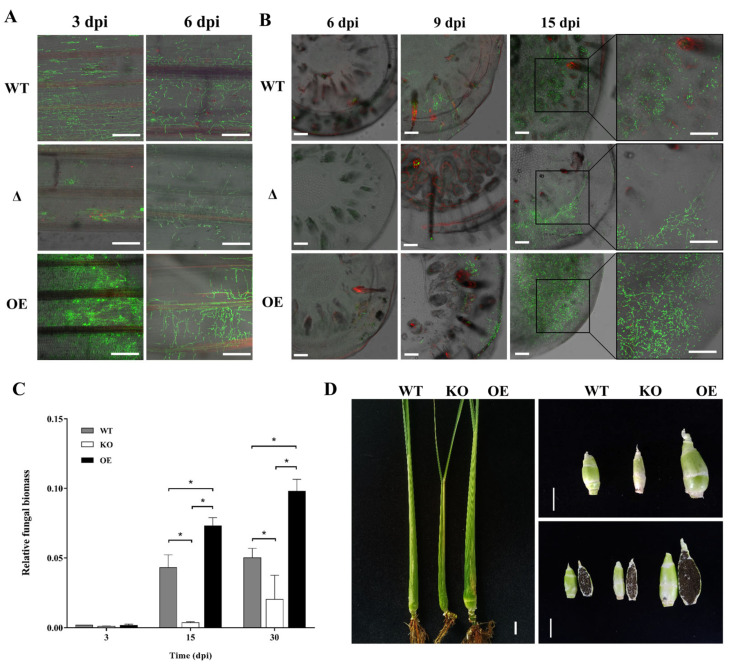
*UeEgl1* helped the proliferation of *U. esculenta* in the host. (**A**,**B**) *Z. latifolia* seedlings inoculated by WT (UeT14 + UeT55), KO (UeT14ΔUeEgl1 + UeT55ΔUeEgl1), OE (UeT14ΔUeEgl1::pHSP-UeEgl1-eGFP + UeT55ΔUeEgl1::pHSP-UeEgl1-eGFP) were observed by confocal microscopy, leaf sheath at 3 and 6 dpi, stem tip at 6, 9 and 15 dpi. sample was stained with WGA/PI. Bar = 150 μm. (**C**) Relative fungal biomass was measured by qPCR at 3, 6, 9, 15, and 30 dpi after inoculation with indicated strains. Relative fungal biomass was represented by the *β-actin* gene of *U. esculenta* normalized to the *β-actin* gene of *Z. latifolia*. “*” indicates significantly higher (Student’s *t*-test, *p* < 0.05). (**D**) Stem swollen state. Image on the left showed swollen *Z. latifolia* plants injected with strains corresponding to the text above, the upper right image showed swollen stems of *Z. latifolia* in the left image, and stems were cut in half to form the image in the lower right corner. Bar = 2 cm.

**Figure 6 jof-08-01050-f006:**
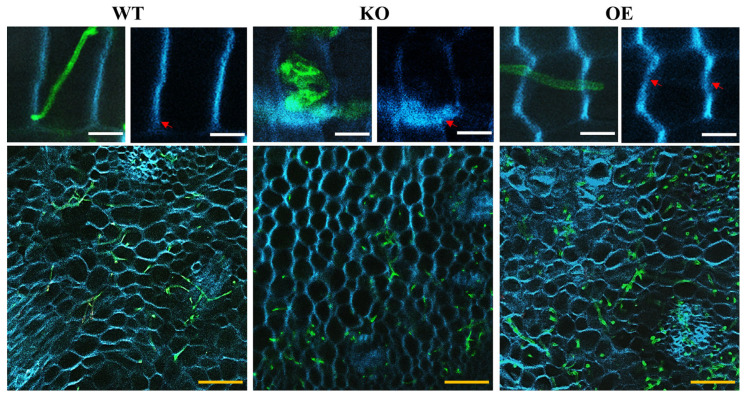
*Z. latifolia* cell wall states were affected by UeEgl1. Images of plants inoculated with WT, KO, and OE at 3 dpi were divided into three sections, the image on the upper left was the leaf sheath of merged bright field and WGA/CFW staining tunnel, the upper right image was the leaf sheath of CFW staining tunnel. The plant stem image below was stained with WGA/CFW. White bar = 20 μm. Yellow bar = 100 μm. The staining of WGA (green) showed the infection dynamics of hyphae, while CFW (blue) can bind cellulose and chitosan in the plant cell wall. Red arrows pointed out fluorescent signal changes.

**Figure 7 jof-08-01050-f007:**
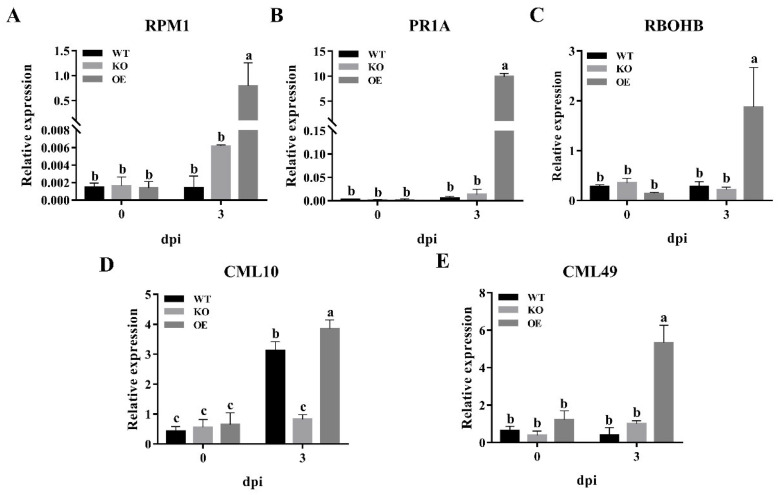
Relative gene expression analysis of specific genes of pathogenesis-related (*PR*) genes, antioxidant enzymes, and calcium-binding proteins genes in *Z. latifolia* inoculated by WT, KO, OE at 0, 3 dpi. Relative expression was represented by qRT-PCR analysis. (**A**) Disease resistance protein (*RPM1*) gene. (**B**) Pathogenesis-related protein 1A (*PR1A*) gene. (**C**) NADPH oxidase (*RBOHB*) gene. (**D**) Calcium-binding protein 10 (*CML10*) gene. (**E**) Calcium-binding protein 49 (*CML49*) gene. Gene relative expression was normalized to the expression of *β-actin*, different letters above the columns are significantly different at the analysis of variance (ANOVA) (*p* < 0.05).

## Data Availability

Not applicable.

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
