# Peer review of "An Endoglucanase Secreted by Ustilago esculenta Promotes Fungal Proliferation"

_jof, 2022, doi:10.3390/jof8101050_

Round 1

Reviewer 1 Report

The manuscript authored by Zihong Ye and collegues describes the identification of endoglucanase egl1 which is differentally regulated in the two infection forms of Ustilago esculenta. While it is upregulated during infection of the T-type it is not in infections with the MT-type which is the more tamed edible form. The authors conduct a solid characterization of an overexpression and a deletion mutant of egl1 including the study of effects on yeast-like growth and on infection. They observe reduced disease symptoms in the deletion mutant while the overexpressor showed stronger symptoms. It was a bit unexpected, that at the same time the plant defence responses of the plant were enhanced in the overexpressor infections.

 One major issue is that I do not believe that U. esculenta enters the cytoplasm of the host cells. Usually, biotrophs break through the cell wall and then enterplant cells without destroying them, that is, they invaginate the plasma membranes for growing through the cell without direct contact to the cytoplasm. Please check on that and adapt the text accordingly. 

Besides that I think the paper is solid and interesting and can be published after correction of few minor things and English corrections:

1) Abstract:  Line 10: provide full genus of Z. latifolia

2) Line 43: Please clarify the issue with the cytoplasm

3) Line 58: no correct sentence

4) The authors barely provide references in their materials and methods although I guess most methods are published or adapted from U. maydis

5) Figure S1: A scheme of the genetic setup would be helpful to understand the PCR results

6) Figure S1C: the panels are too small to see the localisation of egl1-Gfp correctly. Please provide magnifications. As it is a secreted protein by prediction, one would expect it in the vicinity of the cell wall or in the endomembrane pathway... 

7) Font in the figures differs eg in figure 1

8) Line 236: is this analysis published, then please provide a reference 

9) Lines 265 - 277: the figure has two legends, please check. Figure S3 is lacking I guess and figure numbering has to be checked. The legends also differ in their content although I guess they describe the identical figure. 

10) Figure 2A: Mention drawing is not to scale

11) Figure 4B: Cell in the overexpressor look elongated? 

12) Line 402: see my comments above on entering of cytoplasm

Author Response

RESPONSE TO REVIEWS

Reviewer(s)' Comments to Author:
Reviewer: 1
Comments and Suggestions for Authors

The manuscript authored by Zihong Ye and collegues describes the identification of endoglucanase egl1 which is differentially regulated in the two infection forms of Ustilago esculenta. While it is upregulated during infection of the T-type it is not in infections with the MT-type which is the more tamed edible form. The authors conduct a solid characterization of an overexpression and a deletion mutant of egl1 including the study of effects on yeast-like growth and on infection. They observe reduced disease symptoms in the deletion mutant while the overexpressor showed stronger symptoms. It was a bit unexpected, that at the same time the plant defense responses of the plant were enhanced in the overexpressor infections.

One major issue is that I do not believe that U. esculenta enters the cytoplasm of the host cells. Usually, biotrophs break through the cell wall and then enter plant cells without destroying them, that is, they invaginate the plasma membranes for growing through the cell without direct contact to the cytoplasm. Please check on that and adapt the text accordingly.

Besides that I think the paper is solid and interesting and can be published after correction of few minor things and English corrections:

1) Abstract: Line 10: provide full genus of Z. latifolia

Author response: Thanks very much for your valuable suggestions. Zizania latifolia belongs to the tribe Oryzeae (Poaceae). We added the relevant information in the newly submitted manuscript.

2) Line 43: Please clarify the issue with the cytoplasm

Author response: U. esculenta usually proliferates between the plant cell wall and plasma membrane as you described, we misused the term. We have revised “cytoplasm” to “cell wall” in the newly submitted manuscript.

3) Line 58: no correct sentence

Author response: We revised this sentence in newly submitted manuscript.

4) The authors barely provide references in their materials and methods although I guess most methods are published or adapted from U. maydis

Author response: Most our materials and methods are published and cited in this manuscript, and are adapted from U. maydis.

5) Figure S1: A scheme of the genetic setup would be helpful to understand the PCR results

Author response: We have added a scheme as suggested.

6) Figure S1C: the panels are too small to see the localisation of egl1-Gfp correctly. Please provide magnifications. As it is a secreted protein by prediction, one would expect it in the vicinity of the cell wall or in the endomembrane pathway...

Author response: We have rearranged Figure S1 to better visualize the eGFP fluorescent signal.

8) Line 236: is this analysis published, then please provide a reference

Author response: This analysis have not published yet, we will upload the result soon.

9) Lines 265 - 277: the figure has two legends, please check. Figure S3 is lacking I guess and figure numbering has to be checked. The legends also differ in their content although I guess they describe the identical figure.

Author response: Figure S3 is lost, it is the expression patterns of UeEgl1 of MT-type during development stages. We have added the picture in newly submitted manuscript.

10) Figure 2A: Mention drawing is not to scale

Author response: We have revised the picture as suggested.

11) Figure 4B: Cell in the overexpressor look elongated?

Author response: Cell length between wild-type and overexpressor have no significant difference, morphology picture of overexpressor that we have changed are other part of the same picture, and there are no significant change in cell length compare to wild-type.

12) Line 402: see my comments above on entering of cytoplasm

Author response: We have revised “cytoplasm” to “cell wall” in the newly submitted manuscript.

Reviewer 2 Report

Firstly, I would like to congratulate the authors on their nicely prepared and executed research. However, I have some objections over the manuscript, mainly the language used and numerous typos in the text- I believe these can be corrected and polished after a thorough reading through the manuscript. Also, the presented images and figures are not okay, please see my suggestions below:

Presented figures and images should also be modified/clarified:

Figure 1. PLease correct what the relative expression means- relative to what, what is the unit? Percentage? PLease be specific.

Figure 2- instead of B&W please put a colored image

in figure 3- relative enzyme activity- relative to what should be specified on the graph.  Please correct it! WT, KO, OE- it should be explained behind the image

figure 4- images should be larger, it will make them clear

figure 5- relative fungal biomass- what unit? Please correct the graph. Images A and B are too dark- please correct them.

e.g. figure 6- please use an arrow to point to the observed changes

figure S4.B- the heatmap should be larger, otherwise does not make sense

figure s5- this is important, it should be larger, use bigger font too, this way is not clear

figure 7- in what unit is relative expression presented? Relative to what? Please explain on the graph

Please use a separate subtitle for the conclusion and be clear about what have you concluded from the results, was the hypothesis met, and what is the merit/impact of this research for further investigation(s).

Author Response

RESPONSE TO REVIEWS

Reviewer(s)' Comments to Author: 

Reviewer: 2
Comments and Suggestions for Authors

Firstly, I would like to congratulate the authors on their nicely prepared and executed research. However, I have some objections over the manuscript, mainly the language used and numerous typos in the text- I believe these can be corrected and polished after a thorough reading through the manuscript. Also, the presented images and figures are not okay, please see my suggestions below:

Presented figures and images should also be modified/clarified:

Figure 1. PLease correct what the relative expression means- relative to what, what is the unit? Percentage? PLease be specific.

Author response: Thanks very much for your valuable suggestions. Relative expression means the expression of UeEgl1 related to the expression of β-actin, the value of ordinate is a relative value.

Figure 2- instead of B&W please put a colored image

Author response: We have revised the picture as suggested.

in figure 3- relative enzyme activity- relative to what should be specified on the graph.  Please correct it! WT, KO, OE- it should be explained behind the image

Author response: As the legend of Figure 3 shows that endoglucanase activity was normalized to that of the UeT14 strain. WT, KO, OE were explained as suggested.

figure 4- images should be larger, it will make them clear

Author response: We have revised the picture as suggested.

figure 5- relative fungal biomass- what unit? Please correct the graph. Images A and B are too dark- please correct them.

Author response: Relative fungal biomass was represented by the β-actin gene of U. esculenta normalized to the β-actin gene of Z. latifolia. We have revised as suggested and detailed description in the method.

e.g. figure 6- please use an arrow to point to the observed changes

Author response: We have revised as suggested.

figure S4.B- the heatmap should be larger, otherwise does not make sense

Author response: We have revised as suggested.

figure s5- this is important, it should be larger, use bigger font too, this way is not clear

Author response: We have revised as suggested.

figure 7- in what unit is relative expression presented? Relative to what? Please explain on the graph

Author response: Gene relative expression was related to the expression of β-actin of Z. latifolia, the unit was the ratio of gene CT value to β-actin CT value. We have revised as suggested.

Please use a separate subtitle for the conclusion and be clear about what have you concluded from the results, was the hypothesis met, and what is the merit/impact of this research for further investigation(s).

Author response: We have revised as suggested, we summarized our result in the first paragraph of conclusion, and we discussed the impact and new method of this research for further investigations.
